# Association of HTTLPR, BDNF, and FTO Genetic Variants with Completed Suicide in Slovakia

**DOI:** 10.3390/jpm13030501

**Published:** 2023-03-10

**Authors:** Aneta Bednarova, Viera Habalova, Silvia Farkasova Iannaccone, Ivan Tkac, Dominika Jarcuskova, Michaela Krivosova, Matteo Marcatili, Natasa Hlavacova

**Affiliations:** 12nd Department of Psychiatry, Faculty of Medicine, Pavol Jozef Safarik University, University Hospital of Louis Pasteur, 041 90 Kosice, Slovakia; 2Department of Medical Biology, Faculty of Medicine, Pavol Jozef Safarik University, 040 11 Kosice, Slovakia; 3Department of Forensic Medicine, Faculty of Medicine, Pavol Jozef Safarik University, 040 11 Kosice, Slovakia; 44th Department of Internal Medicine, Faculty of Medicine, Pavol Jozef Safarik University, University Hospital of Louis Pasteur, 041 90 Kosice, Slovakia; 51st Department of Psychiatry, Faculty of Medicine, Pavol Jozef Safarik University, University Hospital of Louis Pasteur, 040 11 Kosice, Slovakia; 6Biomedical Centre Martin, Jessenius Faculty of Medicine in Martin, Comenius University in Bratislava, 036 01 Martin, Slovakia; 7Department of Mental Health and Addiction, Fondazione IRCCS San Gerardo dei Tintori, 209 00 Monza, Italy; 8Institute of Experimental Endocrinology, Biomedical Research Center, Slovak Academy of Sciences, 845 05 Bratislava, Slovakia

**Keywords:** suicide, polymorphism, BMI, BDNF, rs962369, SLC6A4, HTTLPR, FTO

## Abstract

Since suicide and suicidal behavior are considered highly heritable phenotypes, the identification of genetic markers that can predict suicide risk is a clinically important topic. Several genes studied for possible associations between genetic polymorphisms and suicidal behaviors had mostly inconsistent and contradictory findings. The aim of this case-control study was to evaluate the associations between completed suicide and polymorphisms in genes *BDNF* (rs6265, rs962369), *SLC6A4* (*5-HTTLPR*), and *FTO* (rs9939609) in relation to sex and BMI. We genotyped 119 completed suicide victims and 137 control subjects that were age, sex, and ethnicity matched. A significant association with completed suicide was found for *BDNF* rs962369. This variant could play a role in completed suicide, as individuals with the CC genotype were more often found among suicides than in control subjects. After sex stratification, the association remained significant only in males. A nominally significant association between the gene variant and BMI was observed for *BDNF* rs962369 under the overdominant model. Heterozygotes with the TC genotype showed a lower average BMI than homozygotes with TT or CC genotypes. *FTO* polymorphism (rs9939609) did not affect BMI in the group of Slovak suicide completers, but our findings follow an inverse association between BMI and completed suicide.

## 1. Introduction

Suicide is the cause of more than 700,000 deaths every year [1]. Although the data from 2019 showed that the suicide mortality rate had a decreasing trend (from 19.65 to 10.50 per population of 100,000) in Europe between the years 2000–2019, the COVID-19 pandemic could have affected the current situation, and the most recent data are yet to come [2,3]. In Slovakia, the suicide mortality rate was lately reported to be 9.31 per 100,000 inhabitants, being higher with increasing age. The prevalence of completed suicide is almost seven-fold higher in Slovak men than women [4,5]. 

Suicidal behavior is characterized as suicide ideation, suicide planning and attempts, and suicide itself [6]. The contributing factors are categorized as distal (genetic predisposition, personality, early life trauma, neurobiological disturbances) and proximal, such as psychiatric disorder, physical disorder, and acute psychosocial crisis [7]. Indeed, suicide and suicidal behavior are considered highly heritable phenotypes [8,9]. However, the specific genes involved and underlying mechanisms together with the role of epigenetics are yet to be elucidated. So far, genetic association studies have discovered various candidate genes potentially involved in suicide susceptibility, and among the most studied have been genes affecting the serotoninergic system, other neurotransmitters from the monoaminergic system, or neurotrophic factors [10]. Since the discovery of brain-derived neurotrophic factor (BDNF), it has been widely studied in relation to various neuropsychiatric disorders [11]. The essential role of BDNF in neuronal growth, differentiation, maturation, and synaptic plasticity have led several research groups also to search for an association with suicidal behaviors by measuring its peripheral levels [12], *BDNF* gene expression, and epigenetic changes [13], but also the influence of *BDNF* polymorphisms [14]. A functional polymorphism of the *BDNF* gene, the Val66Met (rs6265) has been relatively widely investigated with regard to suicidal behavior; however, the findings are inconsistent. This single-nucleotide polymorphism (SNP) might increase the risk of suicidal behavior in certain ethnicities such as Caucasians and Asians, but not in the overall population [14]. Among other *BDNF* gene polymorphisms, rs962369 was found to have a strong association with suicidal ideation during antidepressant treatment [15] as well as with suicide completion [16]. We have previously demonstrated a significant association between *BDNF* (rs6265) and highly polygenic region *5-HTTLPR* of the *SLC6A4* (a gene-encoding serotonin transporter) genetic variations in patients suffering from affective disorders [17]. The latter genetic variant has also been studied with suicide. However, a recent meta-analysis did not confirm its association with suicidal behavior [18]. On the other hand, the authors found a significant association with the risk of violent suicide attempts. Another gene that is potentially involved in psychiatric disorders and suicide attempts could be the fat mass and obesity-associated gene (*FTO*), also known as alpha-ketoglutarate-dependent dioxygenase. Although the *FTO* gene was primarily found to be related to obesity and higher BMI (body mass index) risk [19], abundant expression of FTO in the brain [20] led to subsequent association studies with affective disorders [21,22] or completed suicide [23]. The important role of FTO in modulating brain functions was suggested by several studies in animal models [24,25,26]. Additionally, it was demonstrated that FTO is necessary for the correct function of the hippocampus by regulating BDNF processing [27].

The aim of this study was to evaluate the associations between completed suicide and polymorphisms in genes *BDNF* (rs6265, rs962369), *SLC6A4* (*5-HTTLPR*), and *FTO* (rs9939609) in the Slovak population. We put extra attention on the evaluation of these polymorphisms in relation to the sex and BMI of the study subjects.

## 2. Materials and Methods

### 2.1. Study Sample

The current case-control study was performed in two groups of adult participants of Slovak origin (Caucasians). The case (suicide) group (n = 119) comprised individuals who committed suicide. In Slovakia, all deaths due to suicide or suspected suicide are referred to a coroner’s investigation and reported to the police. Suicide belongs to violent death, and it is necessary to order an autopsy according to legislation in Slovakia. Autopsies, including of suicides, were performed at the Medico-Legal and Pathological-Anatomical Department of the Health Care Surveillance Authority in Kosice (one of the seven departments in Slovakia). The control group (n = 137) was matched by age, sex, and ethnicity to the suicide group. The control group included adult volunteers with no relation to the cases, no known psychiatric disorders, and with no history of suicidal behavior. Blood samples from both groups were collected from November 2018 to November 2022, and the participation of control subjects in testing was voluntary and could be canceled by any individual at any time during the study. Classification of the subjects into the categories of underweight, normal weight, overweight, or obese was based on the cut-off values of BMI recommended for white populations [28]. This study was approved by the Ethics Committee of the University Hospital of Louis Pasteur in Kosice (149/2018/OPVaV), and control subjects provided written informed consent. The study was conducted in accordance with the Helsinki Declaration.

### 2.2. Genotyping

DNA from peripheral blood was extracted using the QIAamp DNA Blood Mini QIAcube Kit according to the manufacturer’s instructions on the QIAcube—robotic workstation for automated purification of DNA, RNA, or proteins (QIAGEN, Hilden, Germay). Genotyping of the *FTO* rs9939609, *BDNF* rs6265, and *BDNF* rs962369 was performed using asymmetric (primers ratio 1:10) real-time polymerase chain reaction and subsequent high-resolution melting analysis in the presence of an unlabeled probe on the Eco Real-Time PCR System (Illumina, Inc., San Diego, CA, USA). The reaction mixture contained 1× MeltDoctor™ HRM Master Mix (Applied Biosystems™, Waltham, MA, USA), appropriate oligonucleotides, and 20 ng of template DNA in a final volume of 15 μL. Genotypes were identified using Eco^TM^ Software 4.1. The oligonucleotides were designed in our laboratory (Table 1). The *5-HTTLPR* genetic variation was genotyped using forward 5′-GGCGTTGCCGCTCTGAATGC-3′ and reverse 5′-GAGGGACTGAGCTGGACAACCAC-3′ primers according to Murakami et al. [29]. The PCR products were separated by electrophoresis on a 2% agarose gel.

### 2.3. Statistical Analysis

The Hardy–Weinberg equilibrium (HWE) assumption was assessed for the tested groups by comparing the observed numbers of each genotype with those expected under the HWE for the estimated allele frequency. Online software SNPstats was used to assess the strength of the relative associations via odds ratios (ORs) with their corresponding 95% confidence intervals (CIs) and *p*-value [30]. Codominant, dominant, overdominant, and recessive genetic models were used to analyze the association between genetic variations and phenotypes. Akaike’s Information Criteria (AIK) and Bayesian Information Criteria (BIC) were considered in model selection. Analysis of association was based on linear or logistic regression according to the response variable (quantitative or binary status (case-control), respectively). The Bonferroni correction was applied to correct for multiple testing. Because four polymorphisms were evaluated, a *p*-value of <0.0125 (a nominal value <0.05/4) was considered statistically significant. 

## 3. Results

### 3.1. Characterization of the Subjects

A total of 256 subjects were included in the current study. Out of 119 suicides, 92 (77.3%) were males, and 27 (22.7%) were females (female: male ratio was 1:3.4). The mean age ± standard deviation (SD) of the suicide cases was 47.71 ± 17.57 years: 46.85 ± 16.65 in males and 50.63 ± 20.11 in females (Table 2). Obtained data showed seven categories of suicide methods. The most common suicide method was hanging or suffocation (71 persons, 59.66%), jumping (16 persons; 13.45%), and collision with a train (12 persons; 10.08%). The remaining suicide methods were categorized as shooting (9%), cutting (4%), and less than 2% frequency for electrocution or drug poisoning. Overall, 98% of cases completed suicide with the violent method. Suicide methods’ frequencies and BMI of suicide cases are shown in Table 3. Age-, gender-, and ethnicity-matched adults served as the control group. The mean age ± SD of the control group was 48.23 ± 18.85 years. Of 137 individuals, 107 (78.1%) were males, and 30 (21.9%) were females. The mean age was 47.36 ± 18.75 years in males and 51.33 ± 18.86 years in females (Table 2).

All collected blood samples were successfully genotyped for all genetic variations. The genotype distribution among controls and cases did not deviate from HWE (*p* ≥ 0.05) for all tested genetic variations.

### 3.2. Distribution of Genotypes of Selected Genetic Variants

As not all analyzed genetic models were statistically significant for *5-HTTLPR* (ins/del), *FTO* (rs9939609), and *BDNF* (rs6265) variants, the distribution of genotypes and their possible association with completed suicide are presented under a codominant genetic model. In the case of the *BDNF* (rs962369), the results are also presented using a recessive model, since it achieved lower AIK and BIC scores than the co-dominant model (Table 4). 

The distribution of *5-HTTLPR* genotypes was similar between controls and cases (Table 4). There were no significant differences in male or female groups when evaluated separately. There was a trend for a higher frequency of the SS genotype in female suicide completers in comparison with the control female group (37% vs. 16.7%); however, the difference did not reach statistical significance. 

No statistically significant association was found between *FTO* or *BDNF* (rs6265) and completed suicide under four studied genetic models nor after sex stratification (data are shown only for the codominant model in Table 4 and Table 5, respectively). We have, however, revealed a nominally significant increased risk of suicide in the subjects with the CC genotype in *BDNF* (rs962369) using a recessive model (CC vs. TT + TC OR = 3.39, 95% CI = 1.05–10.94, *p* = 0.03), irrespective of sex (Table 5). Having considered sex differences, a significant association between *BDNF* rs962369 and suicide was found only in males under the recessive model (CC vs. TT+TC; OR = 4.71, 95% CI = 1.27–17.43, *p* = 0.01). The results show that 12% of male suicide completers had the CC genotype, whereas in the female suicides, we did not find any individual with the CC genotype out of 27 suicide completers. 

### 3.3. Association between HTTLPR, FTO, and BDNF Gene Variants and BMI in Suicide Completers

BMI values of 59.6% of suicide completers were within the normal range (18.5–25), 27.73% were overweight, 5% of subjects were underweight, and 7.6% were obese. The average BMI in all genotype categories ranged from 22.68–25.27 (Table 6). We assessed the association between four gene variants and the BMI of the suicide completers. A nominally significant association between the gene variant and BMI was observed for *BDNF* rs962369 under the overdominant model (*p* = 0.02). Heterozygotes with the TC genotype showed a lower average BMI of about 1.69 (95% C.I. −3.09–−0.29) than homozygotes with TT or CC genotypes. No statistically significant association between *HTTLPR* (ins/del), *FTO* rs9939609, or *BDNF* (rs6265) genotypes and BMI was found either in the codominant or in the dominant, recessive, or overdominant genetic models.

## 4. Discussion

Here, we present an association study of selected gene polymorphisms with suicide completion within the Slovak population. We investigated a serotonin system candidate, a functional polymorphism *5-HTTLPR* (ins/del) in the *SLC6A4* gene, two common polymorphisms in the widely studied neurotrophic factor gene *BDNF* (rs6265, rs962369), and finally, an *FTO* polymorphism (rs9939609). This is the first study exploring the risk of committing completed suicide in the Slovak population related to these selected variants. In our study, the female-to-male ratio in the case subjects was 1:3.4, which is in line with the sex discrepancy in suicide rates in Slovakia [4,5]. It is estimated that around 20% of global suicides are due to pesticide self-poisoning (nonviolent method), most of which occur in rural agricultural areas in low- and middle-income countries. Other common methods of suicide are hanging and firearms [1]. In our study, 98% of cases overall completed suicide with the violent method; the most frequent method (around 60%) was hanging or suffocation, a method with an increasing trend in many countries over the last 30 years [31,32].

### 4.1. Evaluation of an Association of BDNF Genetic Variants with Suicide

A promising candidate for an association with suicidal behavior is the *BDNF* gene. Two SNPs previously associated with suicidal behavior were analyzed in our study. The most studied *BDNF* polymorphism is rs6265 (Val66Met). In our study, we did not find significant differences in rs6265 under all analyzed genetic models, and no differences were found after gender stratification. Our results are in accordance with a meta-analysis study in which a significant association between *BDNF* rs6265 polymorphism and the risk of suicidal behavior in the overall population was not confirmed [14]. This meta-analysis included studies from three populations of Slavic origin (Croatia, Slovenia, and Poland), out of which three did not confirm any association [33,34,35], and one found a higher risk for carriers of at least one minor allele (Met-Met or Val-Met) but only in female suicide victims [36]. Neither the more recent study from Central Europe [37] nor an earlier European multicenter study [38] has found an association between rs6365 and suicidal behavior.

Although this broadly studied missense variant of *BDNF* did not show significant results, another polymorphism, rs962369, could be another candidate for predicting increased suicide risk. In our study, out of four observed polymorphisms, the only association with completed suicide was found for *BDNF* rs962369. The association with *BDNF* rs962369 and susceptibility to suicide has been significant after gender stratification in males under recessive models. On the contrary, in the recent Slovenian study, none of the seven BDNF polymorphisms showed an association with completed suicide when used as a single marker. However, haplotype analysis of five selected polymorphisms showed that in such a combination, major allele T of rs962369 contributed to a higher suicide risk association [16]. Out of 123 polymorphisms in the GENDEP project, the strongest association with suicidal ideation was observed in rs962369 genetic variants [15]. Although the study assessed only suicidal ideation during antidepressant treatment, the outcomes correspond with our observation. Studies on animal models show the tendency of developing BDNF-deficient-related diseases such as depression or anxiety is higher in female animals [39], and that sex hormones or steroids can modulate the activities of BDNF, which may account for its functional discrepancy in different sexes [40,41]. Although the number of female cases and controls in our study was too low to make any conclusions, we speculate that the *BDNF* rs962369 variant could act in a sex-dependent manner.

### 4.2. Evaluation of an Association of 5-HTTLPR (ins/del) Genetic Variants with Suicide

One of the most studied genetic variations associated with suicide behavior is a highly polymorphic region *5-HTTLPR* (ins/del) of the *SLC6A4* gene-encoding serotonin transporter. So far, controversial results have been obtained. Out of systematic reviews and meta-analyses, some studies confirmed the higher susceptibility of short allele (S) carriers for suicidal behavior [42,43,44], but others have not [18,45], or contradictory results have been found in which a positive association was found with long allele (L) carriers [46]. Interestingly, discrepancies have also been found when distinct phenotypes such as suicidal behavior generally, suicidal ideation, history of suicide attempts, or a complete suicide only are considered. The 5-HTTLPR polymorphism has been found to be significantly associated with suicide attempts, but not with completed suicides [42,45]. Other meta-analyses did not confirm the association of the S allele and suicidal behavior generally; however, they found an association of the S allele with suicide attempts within the same psychiatric diagnosis, also with violent suicide [47] or violent suicide attempt [18].

In the present study, we did not reveal any significant difference between genetic variations of the 5-HTT gene promoter comparing suicide cases and controls. The result follows the studies performed in other Central Europe populations [48,49] and within individuals of Slavic origin [50,51,52]. Interestingly, a huge study that involved more than 100,000 subjects from 38 countries investigating an association between allelic frequencies in various countries and their suicide rates found that the S allele acts as a protective factor in Caucasians, whereas it acts as a risk factor in non-Caucasian populations [53]. We also evaluated *HTTLPR* genetic variants after gender stratification, as some studies found an association only in males [54] and others only in females [55]. Our results did not show either of these associations, although there was a trend for a higher frequency of the SS genotype in female suicide completers.

### 4.3. The FTO Genetic Variant (rs9939609) in the Context of Suicide and BMI

Within the genes related to obesity, the *FTO* gene has one of the strongest links with this condition in the human population. In the most studied polymorphism of this gene, rs9939609, minor allele A was found to positively affect obesity and BMI [19,56]. The *FTO* gene has recently been studied in association with affective disorders given its high abundance in the brain and often comorbid obesity and depression. A bidirectional relationship between obesity and higher BMI with depression has been proposed lately [57], although the underlying mechanisms have not yet been elucidated [58]. Up to 12% of a shared genetic component was found between depression and obesity [59]. However, there are various phenotypes within depressive disorder, and though one type is characterized by hyperphagia, another can lead to weight loss [60]. The meta-analysis by Rivera et al. [61] supported a significant interaction between *FTO*, depression, and BMI, indicating that depression increases the effect of *FTO* on BMI; depressed subjects had an additional effect of *FTO* on BMI that corresponded to a BMI increase of 2.2% for each A allele. The mentioned meta-analysis did not confirm an association between the rs9939609 A risk allele and depression, similar to that in the earlier studies [21,22]. Considering the association studies between this polymorphism and suicide, an inverse association between the A allele and complete suicide was reported in the Polish population, suggesting a codominant effect of the risk allele [23]. One study has suggested that the A allele could be a protective variant for depression development [62].

We aimed to test the correlation between the BMI and FTO rs9939609 genotypes in a group of suicide completers. In our study, we did not reveal any significant association between *FTO* gene polymorphism and BMI in suicide completers, even after gender stratification.

This study has limitations such as its relatively small number of cases and controls, particularly in the female groups. However, this corresponds with the discrepancy in suicide rate frequency in women and men in the Slovak population. The study’s strength is that the control subjects were cautiously matched by age, gender, and ethnicity to their suicide counterparts. Another strength is that this is the first Slovak association study of variants 5-*HTTLPR* (ins/del) in the *SLC6A4* gene, *BDNF* (rs6265, rs962369), and *FTO* (rs9939609) with completed suicide. Our main result is that the *BDNF* rs962369 variant could play a role in completed suicide, as individuals with the CC genotype were more often found among suicides than in control subjects. The finding was confirmed mainly for male individuals. In the female group, the result cannot be interpreted unequivocally due to the small number of cases and controls. Another important finding of this study is that *FTO* polymorphism (rs9939609) did not affect BMI in the group of Slovak suicide completers, but our findings follow an inverse association between BMI and completed suicide.

## 5. Conclusions

The results of this study suggest that rs962369 polymorphism of the *BDNF* gene could potentially be involved in the higher risk of committing complete suicide in the Slovak male population. We did not find any association between other evaluated genetic variants, but studies with more included subjects are needed to verify this finding. Interestingly, *FTO* genetic variants did not affect the BMI of Slovak suicide completers.

## Figures and Tables

**Table 1 jpm-13-00501-t001:** Sequences of oligonucleotides.

Gene	Oligonucleotides	Sequences
*FTO* rs9939609	forward-limit	5′-GCATTTAGAATGTCTGAATTATTATTCTAG-3′
	reverse-excess	5′-CCTATTAAAACTTTAGAGTAACAGAG-3′
	probe	5′-TGCTGTGAATTTTGTGATGCACTTGGAT-Phos′
*BDNF* rs6265	forward-limit	5′-GCCGAACTTTCTGGTCCTCATCC-3′
	reverse-excess	5′-AAGGCAGGTTCAAGAGGCTTG-3
	probe	5′-GCTCTTCTATCACGTGTTCGAAAGTGTC-Phos
*BDNF* rs962369	forward-limit	5′-GACATTTTTATGAGAAGGGTTTACATAAG-3′
	reverse-excess	5′-AAAGAATTGCTCACTGTAATGAC-3′
	probe	5′-TGCCAAGAGAGTTGAGTCCATGG-Phos

**Table 2 jpm-13-00501-t002:** Age characteristics of control and suicide group of the subjects.

Age	Control			Suicides		
	Total (n = 137)	Males (n = 107)	Females (n = 30)	Total (n = 119)	Males (n = 92)	Females (n = 27)
average ± SD	48.23 ± 18.85	47.36 ± 18.75	51.33 ± 18.86	47.71 ± 17.57	46.86 ± 16.65	50.63 ± 20.11
median	47	45	52.5	48	46.5	51
min–max	18–91	21–90	18–91	14–90	14–85	15–90
<35 years	40 (29.20%)	34 (31.78%)	6 (20.00%)	31 (26.05%)	26 (28.26%)	5 (18.52%)
35–49 years	30 (21.90%)	25 (23.36%)	5 (16.67%)	35 (29.41%)	29 (31.52%)	6 (22.22%)
>50 years	67 (48.91%)	48 (44.86%)	19 (63.33%)	53 (44.54%)	37 (40.22%)	16 (59.26%)

Data are expressed as mean ± standard deviation (SD)/min/max/%.

**Table 3 jpm-13-00501-t003:** Methods of suicide and BMI in suicide completers.

	Total		Males		Females	
Method of Suicide	n	%	n	%	n	%
hanging or suffocation	71	59.6%	58	61.1%	13	48.2%
jumping from high places	16	13.4%	9	9.47%	7	25.9%
collision with a train	12	10.1%	7	7.37%	5	18.5%
shooting	11	9.24%	11	11.6%	0	0.00%
cutting	5	4.20%	5	5.26%	0	0.00%
electrocution	2	1.68%	2	2.11%	0	0.00%
poisoning	2	1.68%	0	0.00%	2	7.41%
**BMI**						
average ± SD	24.22 ± 3.83		24.17 ± 3.30		24.42 ± 5.35	
median	23.84		23.86		23.53	
min–max	13.6–35.7		13.9–33.31		13.67–35.7	
BMI < 18.5 (underweight)	6	5.04%	3	3.26%	3	11.1%
BMI = 18.5–24.9 (normal)	71	59.6%	61	66.3%	10	37.4%
BMI = 25–29.9 (overweight)	3	27.7%	22	23.9%	11	40.7%
BMI > 30 (obese)	9	7.56%	6	6.52%	3	11.1%

**Table 4 jpm-13-00501-t004:** Distribution of genotypes and analysis of the association between *5-HTTLPR* (ins/del) and *FTO* (rs9939609) gene variants and suicide risk.

Subjects	Genotype/Allele	Control n (%)	Suicides n (%)	Odds Ratio (95% CI)	*p*-Value	HWE Controls
*5-HTTLPR* (ins/del)						
Codominant model						
Total	LL	32 (23.4%)	19 (16.0%)	1.00	0.32	0.94
	LS	68 (49.6%)	63 (52.9%)	1.56 (0.80–3.03)		
	SS	37 (27.0%)	37 (31.1%)	1.68 (0.81–3.49)		
	Allele, S (proportion)	142 (0.52)	137 (0.58)			
Males	LL	25 (23.4%)	17 (18.5%)	1.00	0.65	0.53
	LS	50 (46.7%)	48 (52.2%)	1.41 (0.68–2.94)		
	SS	32 (29.9%)	27 (29.4%)	1.24 (0.56–2.77)		
	Allele, S (proportion)	114 (0.53)	102 (0.55)			
Females	LL	7 (23.3%)	2 (7.4%)	1.00	0.09	0.26
	LS	18 (60.0%)	15 (55.6%)	2.92 (0.53–16.19)		
	SS	5 (16.7%)	10 (37.0%)	7.00 (1.04–46.94)		
	Allele, S (proportion)	28 (0.47)	35 (0.65)			
*FTO* (rs9939609)						
Codominant model						
Total	TT	42 (30.7%)	46 (38.7%)	1.00	0.40	0.61
	TA	65 (47.5%)	49 (41.2%)	0.69 (0.39–1.20)		
	AA	30 (21.9%)	24 (20.2%)	0.73 (0.37–1.44)		
	Allele, A (proportion)	125 (0.46)	97 (0.41)			
Males	TT	34 (31.8%)	36 (39.1%)	1.00	0.55	0.89
	TA	52 (48.6%)	39 (42.4%)	0.71 (0.38–1.32)		
	AA	21 (19.6%)	17 (18.5%)	0.76 (0.35–1.69)		
	Allele, A (proportion)	94 (0.44)	73 (0.40)			
Females	TT	8 (26.7%)	10 (37.0%)	1.00	0.70	0.47
	TA	13 (43.3%)	10 (37.0%)	0.62 (0.18–2.13)		
	AA	9 (30%)	7 (26.0%)	0.62 (0.16–2.42)		
	Allele, A (proportion)	31 (0.52)	24 (0.44)			

**Table 5 jpm-13-00501-t005:** Distribution of genotypes and analysis of the association between *BDNF* (rs6265) and *BDNF* (rs962369) gene variants and suicide risk.

Subjects	Genotype/Allele	Control n (%)	Suicides n (%)	Odds Ratio (95% CI)	*p*-Value	HWE Controls
*BDNF* (rs6265)						
Codominant model						
Total	CC	92 (67.2%)	82 (68.9%)	1.00	0.92	0.82
	CT	41 (29.9%)	33 (27.7%)	0.90 (0.52–1.56)		
	TT	4 (2.9%)	4 (3.4%)	1.12 (0.27–4.63)		
	Allele, T (proportion)	49 (18%)	41 (17%)			
Males	CC	75 (70.1%)	63 (68.5%)	1.00	0.84	0.92
	CT	29 (27.1%)	25 (27.2%)	1.03 (0.55–1.93)		
	TT	3 (2.8%)	4 (4.3%)	1.59 (0.34–7.36)		
	Allele, T (proportion)	35 (0.16)	33 (0.18)			
Females	CC	17 (56.7%)	19 (70.4%)	1.00	0.34	0.52
	CT	12 (40.0%)	8 (29.6%)	0.60 (0.20–1.81)		
	TT	1 (3.3%)	0 (0.0%)	0.00 (0.00–NA)		
	Allele, T (proportion)	14 (0.23)	8 (0.15)			
*BDNF* (rs962369)						
Codominant model						
Total	TT	81 (59.1%)	67 (56.3%)	1.00	0.09	0.20
	TC	52 (38.0%)	41 (34.5%)	0.95 (0.56–1.60)		
	CC	4 (2.9%)	11 (9.2%)	3.32 (1.01–10.92)		
	Allele, C (proportion)	60 (0.22)	63 (0.26)			
Males	TT	65 (60.8%)	51 (55.4%)	1.00	**0.04**	
	TC	39 (36.5%)	30 (32.6%)	0.98 (0.54–1.79)		
	CC	3 (2.8%)	11 (12.0%)	**4.67 (1.24–17.64)**		
	Allele, C (proportion)	97 (0.24)	45 (0.21)	52 (0.28)		
Females	TT	16 (53.3%)	16 (59.3%)	1.00	0.50	0.39
	TC	13 (43.3%)	11 (40.7%)	0.85 (0.29–2.44)		
	CC	1 (3.3%)	0 (0.0%)	0.00 (0.00-NA)		
	Allele, C (proportion)	15 (0.25)	11 (0.2)			
Recessive model						
Total	TT+TC	133 (97.1%)	108 (90.8%)	1.00	**0.03**	
	CC	4 (2.9%)	11 (9.2%)	**3.39 (1.05–10.94)**		
Males	TT+TC	104 (97.2%)	81 (88.0%)	1.00	**0.01**	
	CC	3 (2.8%)	11 (12.0%)	**4.71 (1.27–17.43)**		

**Table 6 jpm-13-00501-t006:** Analysis of the association between four gene variants and BMI in suicide cases.

Gene	Model	Gender	Genotype	BMI mean (s.e.)	Difference (95% CI)	*p*-Value
** *HTTLPR* **						
**ins/del**	Codominant	Total	LL	23.93 (0.67)	ref.	0.90
			LS	24.28 (0.47)	0.35 (−1.20–1.90)	
			SS	24.05 (0.82)	0.12 (−1.99–2.24)	
		Male	LL	24.10 (0.50)	ref.	1.00
			LS	24.03 (0.51)	−0.07 (−1.64–1.50)	
			SS	24.06 (0.90)	−0.04 (−2.06–1.98)	
		Female	LL	23.47 (2.14)	ref.	0.76
			LS	25.08 (1.13)	1.61 (−2.68–5.90)	
			SS	23.96 (2.16)	0.49 (−7.65–8.63)	
** *FTO* **						
**rs9939609**	Codominant	Total	TT	23.88 (0.46)	ref.	0.65
			TA	24.53 (0.62)	0.65 (−0.88–2.18)	
			AA	23.84 (0.80)	−0.03 (−1.91–1.85)	
		Male	TT	24.20 (0.44)	ref.	0.71
			TA	24.19 (0.61)	−0.02 (−1.52–1.49)	
			AA	23.46 (0.85)	−0.74 (−2.66–1.17)	
		Female	TT	22.69 (1.43)	ref.	0.41
			TA	25.85 (1.90)	3.15 (−1.43–7.73)	
			AA	24.78 (1.86)	2.08 (−2.96–7.13)	
** *BDNF* **						
**rs6265**	Codominant	Total	CC	24.30 (0.40)	ref.	0.55
			CT	23.59 (0.74)	−0.71 (−2.25–0.83)	
			TT	25.27 (2.04)	0.96 (−2.85–4.78)	
		Male	CC	24.05 (0.41)	ref.	0.74
			CT	23.88 (0.68)	−0.17 (−1.71–1.37)	
			TT	25.27 (2.04)	1.21 (−2.14–4.57)	
		Female	CC	25.13 (1.07)	ref.	0.27
			CT	22.68 (2.24)	−2.45 (−6.73–1.83)	
			TT	NA	-	
** *BDNF* **						
**rs962369**	Codominant	Total	TT	24.65 (0.43)	ref.	0.20
			TC	23.30 (0.66)	−1.35 (−2.81–0.12)	
			CC	24.14 (1.04)	−0.51 (−2.92–1.89)	
		Male	TT	24.71 (0.46)	ref.	0.06
			TC	22.92 (0.57)	**−1.79 (−3.25–−0.33)**	
			CC	24.14 (1.04)	−0.57 (−2.68–1.53)	
		Female	TT	24.45 (1.07)	ref.	0.96
			TC	24.34 (1.98)	−0.11 (−4.19–3.97)	
			CC	NA	-	
	Overdominant	Male	TT and CC	24.61 (0.42)	ref.	**0.02**
			TC	22.92 (0.57)	**−1.69 (−3.09–−0.29)**	

## Data Availability

The data that supports the findings of this study are available upon request.

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
