# Peer review of "Association of HTTLPR, BDNF, and FTO Genetic Variants with Completed Suicide in Slovakia"

_jpm, 2023, doi:10.3390/jpm13030501_

Round 1

Reviewer 1 Report

The study is interesting, I have only two comments. I suggest to verify the results by the correction for multiple testing (for BDNF two polymorphisms were tested). The second comment is on the group of persons who committed suicide. Is there any information on the presence of mental disorders in this group? It would be interesting to add it.

Author Response

Response to reviewers‘ comments

We thank the reviewers for their valuable comments which helped us to improve the manuscript’s quality. The changes made in the revised version of the manuscript are highlighted by track changes in one version of the revised manuscript.

Reviewer 1

The study is interesting, I have only two comments. I suggest to verify the results by the correction for multiple testing (for BDNF two polymorphisms were tested).

Response: We thank the reviewer for this suggestion. We performed Bonferroni’s correction taking into account that four polymorphisms were tested. Thus, the corrected p-value is p<0.05/4 = p<0.0125. This statement is now included in Methods. Corresponding changes were done in the text. The associations which do not fulfill this criterion are referred to as „nominally significant“ while the associations fulfilling corrected p-value are referred to as „statistically significant“.

The second comment is on the group of persons who committed suicide. Is there any information on the presence of mental disorders in this group? It would be interesting to add it.

Response: Thank you for your comment, we do not have this data. They are absent in autopsy reports.

Reviewer 2 Report

In this work, authors evaluate the potential influence of some known genetic alterations with suicidal behavior and with BMI variation in suicide completers. Analyses were performed on a novel sample (adult partecipants who committed suicide and age/sex -matched control) recruited by authors, with the advantage of a more similar genetic background (all partecipants are of Slovakian origin). In this sample authors found a potential association of rs962369 (BDNF gene) with suicide completion.

There are some points which should be considered:

- Authors performed several tests on specific polymorphisms under different assumptions (Codominance/Dominance/Overdominance models) to provide a better picture of the effects of the genetic alterations on the investigated phenotypes. Given the number of test performed, statistical correction should be included in the study (i.e. Bonferroni, FDR). In statistical analysis paragraph - method section, there is no mention to statistical correction. 

- in the last part of discussion section and in the conclusion section, authors state that their findings "are in agreement with the inverse association between BMI and completed suicide". Could authors better explain/discuss this statement, as i did not find enough data on the paper to support it. 

I'd also like to point to some minor issues:

- If possible, tables should be formatted (or divided) to better visualize data. For example, the lenght of table 4 makes difficult to read the data.

- Also, in Table 3, BMI section, there seems to be a repetition on some of the numbers.

I thank authors for their work.

Author Response

Response to reviewers‘ comments

We thank the reviewers for their valuable comments which helped us to improve the manuscript’s quality. The changes made in the revised version of the manuscript are highlighted by track changes in one version of the revised manuscript.

Reviewer 2

Authors performed several tests on specific polymorphisms under different assumptions (Codominance/Dominance/Overdominance models) to provide a better picture of the effects of the genetic alterations on the investigated phenotypes. Given the number of test performed, statistical correction should be included in the study (i.e. Bonferroni, FDR). In statistical analysis paragraph - method section, there is no mention to statistical correction. 

Response: We thank the reviewer for this comment. Based on this suggestion, we performed Bonferroni’s correction taking into account that four polymorphisms were tested. Thus, the corrected p-value is p<0.05/4 = p<0.0125. This statement is now included in Methods. Corresponding changes were done in the text. The associations which do not fulfill this criterion are referred to as „nominally significant“ while the associations fulfilling corrected p-value are referred to as „statistically significant“.

In the last part of discussion section and in the conclusion section, authors state that their findings "are in agreement with the inverse association between BMI and completed suicide". Could authors better explain/discuss this statement, as i did not find enough data on the paper to support it. 

Response: We agree with the reviewer that this part of discussion was an overstatement based on the comparison of our group of suicide patients (n=119) with the whole Slovak population. We deleted the part of Discussion in question and the last sentence in the Conclusion reads: Interestingly, FTO genetic variants did not affect the BMI of Slovak suicide completers. Also the corresponding references were deleted from the Literature.

I'd also like to point to some minor issues:

- If possible, tables should be formatted (or divided) to better visualize data. For example, the lenght of table 4 makes difficult to read the data.

Response: Based on the reviewer’s suggestion, Table 4 was divided into two sections. Now, the revised version of the manuscript comprises 6 tables.

- Also, in Table 3, BMI section, there seems to be a repetition on some of the numbers.

Response: We apologize for the mistake in Table 3. Thank you for noticing this. The data has been corrected in the revised version of the manuscript.